# Interpretation bias and contamination-based obsessive-compulsive symptoms influence emotional intensity related to disgust and fear

**Jakob Fink-Lamotte[1]\*, Andreas Widmann[2,3], Judith Fader[1], Cornelia Exner[1]**

**1** Clinical Psychology and Psychotherapy, University of Leipzig, Leipzig, Germany, **2** Cognitive and Biological Psychology, University of Leipzig, Leipzig, Germany, **3** Leibniz Institute for Neurobiology, Magdeburg, Germany

\* jakob.fink@uni-leipzig.de

## Abstract

Biased processing of disgust-related stimuli is increasingly discussed in addition to fear-related processing as a maintenance factor for contamination-based obsessive-compulsive disorder (C-OCD). However, the differential impact of fear and disgust on biased processing in C-OCD is not yet completely understood. Because it is difficult to distinguish the two emotions in self-report assessment by directly addressing the specific emotions, a text paragraph-based interpretation bias paradigm was applied to more implicitly assess emotions. For the text-based interpretation bias paradigm, disgust-related, fear-related, disgust-fear-ambiguous and neutral text paragraphs describing everyday life situations were developed and validated in a pre-study ($N$ = 205). Fifty-nine healthy participants watched either disgust- or fear-inducing movies and afterwards rated their experienced emotional response to the text paragraphs. The results show that fear and disgust components of an emotional response to mixed-emotional situations are strongly influenced by the situational context, and across the levels of trait contamination fear people did not differ in their fear experiences to everyday situations (which was overall strong), but in their disgust experiences. These findings highlight the strength of situational context on interpretation bias for mixed-emotional disorders and the important role of disgust for C-OCD.

## Introduction

Disgust and fear are distinct basic human emotions [1,2] and they differ in several aspects (e.g. physiological measures as well as habituation and extinction). However, the two emotions also share many characteristics (e.g. aversive emotion, reinforce appropriate behaviors). Both emotions are organized along a motivational defense system (*defence cascade model*) [3]. This system facilitates the processing of threatening context by gathering information in the early stage and switching to overt behavior in a later stage, when the threat becomes more imminent. Nonetheless, this system needs to recognize which information is threatening and which is not. Because disgust and fear are both aversive emotions, it can be suggested that they are both interpreted as threatening. However, many everyday situations are neither exclusively disgust- nor exclusively fear-related. Therefore, this study aims to better understand the extent

**Data Availability Statement:** Fink, Jakob; Widmann, Andreas; Fader, Judith; Exner, Cornelia (2019), "Interpretation bias and contamination-based obsessive-compulsive symptoms influence

emotional intensity related to disgust and fear",
Mendeley Data, v1 http://dx.doi.org/10.17632/
cw8bp2xvtv.1.

**Funding:** . The author(s) acknowledge support
from the German Research Foundation (DFG) and
Universität Leipzig within the program of Open
Access Publishing.

**Competing interests:** The authors have declared
that no competing interests exist.

to which disgust- and fear-specific state and trait factors influence the interpretation of ambiguous situations.

In order to explain the interpretation of ambiguous situations, Mathews and Mackintosh [4] postulated that in ambiguous situations at least two evaluative systems compete over the interpretation narrative, whereby the system winning the high ground suppresses the other system by offering the fastest interpretation. In line with Blanchette and Richards [5], they proposed that interpretation is neither a fast, automatic bottom-up process nor an arduous reconstitution process. According to this hypothesis, only a small amount of active processing is involved, and therefore ambiguous material is interpreted using easily-accessible information (e.g. emotion-congruent information, previous experience) with greater probability. Taken together, the process of interpretation is highly dependent on the accessibility of the information. This might be relevant for anxiety disorders and obsessive-compulsive disorders (OCD), where vulnerable individuals interpret neutral situations as threatening.

Hirsch et al. [6] proposed that interpretation bias is a maintenance factor of emotional disorders due to cognitive errors (e.g. worry, rumination, maladaptive appraisals) that trigger negative interpretations. In line with the models introduced above, these cognitive errors increase the probability that emotion-congruent and threat-relevant information is more accessible. From a longer-term perspective, these "negative interpretations may lead to further threatening resolutions to subsequent ambiguity" (p. 295). The influence of fear on the interpretation bias has been comprehensively investigated, with several studies showing that people with anxiety disorders or obsessive-compulsive disorder are more sensible to anxiogenic stimuli and experience emotional-ambiguous situations as more threatening compared with healthy controls (interpretation bias: [7,8]; for OCD for review: [9]).

However, patients with the OCD sub-type contamination and washing (C-OCD) experience not only aversive feelings of fear but also disgust, whereby both emotions influence the interpretation bias [10]. It has been more difficult to show the specificity of interpretation biases for disgust-related situations. Two studies [11,12] found that after fear induction, participants had increased fear ratings only for fearful pictures, while after disgust induction they rated disgust-relevant pictures as more fearful. Davey, Macdonald and Brierley [13] found that ambiguous fear-disgust stimuli (homophone words, e.g. dye/die)–as well as fear stimuli–were rated as more fearful after disgust compared with neutral induction. Unfortunately, the study lacked in addition to the disgust and neutral induction also an anxiety induction condition. In order to better understand the which extent fear and disgust reinforce each other in the interpretation bias process, it seems necessary to investigate how the induction of both emotion influences ambiguous fear-disgust stimuli. Furthermore, because no specific disgust outcome was assessed yet, it has been attempted to measure both emotions seperately. A more detailed insight into these processes is highly relevant (a) to increase the knowledge of the maintaining processes of emotional disorders with a multiple emotional basis and (b) to improve the psychotherapy of emotional disorders by targeting disgust and fear with highly emotion-specific techniques.

Therefore, the aim of the study is to understand the extent to which disgust- and fear-specific state factors such as easily-accessible (emotion-congruent) information and trait factors like different degrees of trait contamination fear influence the experience of emotional-ambiguous situations. Emotional-ambiguous situations are modeled as text paragraphs describing ambiguous situations that could elicit fear or disgust depending on the interpretation. Emotional state is modulated by movie-based emotion induction. We are further interested in whether state and trait factors influence the emotional experience of the text vignettes independently or interact with each other. Due to the strong arousal induced by disgust and fear, we assume that individuals would not be able to directly differentiate between the two

emotions of fear and disgust [14]. Therefore, in the present study both emotions were assessed by participants' arousal ratings in response to validated normative prototypical disgust and fear situations. First, we expect an emotion-specific interpretation bias, whereby situational factors (like movie-based emotion induction) bias the emotional experience of emotion-congruent but not emotion-incongruent text paragraphs (hypothesis 1). Second, we predict that the emotion-specific interpretation biases will interact with the magnitude of trait contamination fear, with stronger biases among people with higher trait contamination fear (hypothesis 2). Finally, because research shows differences in habituation to disgust and fear [15], we predict that the emotional response to fear-related text paragraphs habituates more strongly compared with disgust-related text paragraphs between the two blocks (hypothesis 3).

## Material and methods

### Subjects

We calculated a necessary sample size with G*power (V. 3.1) [16] of 60, assuming a medium effect size ($f$ = .25), a power of .95, an $\alpha$ of 0.05 and allowing for a drop-out rate (e.g. due to missing and technical problems) of 20%. In this experiment, 59 (52 females, 7 males) voluntary subjects participated. The participants were on average 21.54 years old (SD = 6.361, range: 18–60 years). All participants had graduated from high school (Abitur), and five participants had already received a college degree. The trait anxiety scores showed low to moderate anxiety in both groups (Fear-group: $M$ = 39.10, $SD$ = 10.68; Disgust-group; $M$ = 33.46, $SD$ = 7.29) and no participant met the criteria for current mental disorders assessed by the instruments described below and no participant was in psychotherapy during the time of the experiment. Seven participants were taking medicine (e.g. birth control), but no psychotropic medication. All were German native speakers aside from two participants, who were also fluent in the German language. All participants were bachelor students at the University of Leipzig and received student class credit for participation and all participants gave their written informed consent and had the autonomy to stop at any point during the study. The study was approved by the local ethics committee of the Department of Medicine, University of Leipzig (329-14-06102014).

### Measures

In this experiment, the Mini-International Neuropsychiatric Interview (MINI) [17,18] was used to screen for symptoms of psychological disorders. The MINI was designed as a brief structured interview for the major axis I psychiatric disorders in DSM-IV. Dimensions of trait contamination fear were assessed by applying the four-point scaled self-rated *Padua Inventory–Palatine Revision* (PI-PR) [19,20]. Only the sub-scale for washing and contamination was assessed in the present study, resulting in the *PI washing score*. The PI-PR has been reported to have an acceptable internal consistency (Cronbach's $\alpha$ > .78). In order to control for depressive symptoms, the second revision of the Beck Depression Inventory (BDI-II) [21] was used. The German translation has a high internal consistency (Cronbach's $\alpha$ > .84). In order to measure trait anxiety, the State-Trait-Anxiety Inventory Trait Scale (STAI-T) [22] was applied. The German translation has a high internal consistency (Cronbach's $\alpha$ > .83).

### Stimuli and materials

Different studies [12,23] have shown that movies induce strong and persistent emotions. For inducing emotions, movies are advantageous compared with other emotional stimuli due to the multimodal, dynamic and non-static features. Therefore, four different movie clips inducing disgust or fear were selected based on prior studies of emotions (see below). All four

selected movie clips were synchronized in the form and substance of the emotional content: the topic of both disgust movie clips was feces and excrements, while the topic of both fear-related movie clips was persecution. The first disgust movie clip was a 1 minute and 18 second part of the movie *Pink Flamingo* by John Waters [24] in which the character Divine eats dog feces. The second disgust clip presented was a 1 minute and 30 second part of the movie *Trainspotting* [25]. Hereby, a man sits on a very dirty and disgusting toilet and tries to extracts a bag of drugs from his anus. Both movie clips have previously been used successfully for the same purpose in other experiments [23,26–28]. The first movie clip presented to induce fear was a 2 minute and 47 second part of *Marathon Man* [29] where a person is hunted by agents. The second movie clip presented to induce fear was a 4 minute part of the horror movie *Helloween* [30], in which a babysitter is hunted in her neighbor's house. Both fear movie clips have previously been used successfully in different experiments to induce fear [28,31,32].

The central dependent variable was emotional intensity as experienced in response to text paragraphs of ambiguous content. The text paragraphs were created in line with the procedure of Eysenck et al. [33]. For the present experiment, 28 small text paragraphs were written by the authors, comprising one or two sentences describing everyday life situations, which are suitable to produce an emotional response in most people. The paragraphs were attributable to four categories (see S1 Appendix). The first category of paragraphs contained disgust-related situations (e.g. "You go to a public restroom. Afterwards you realize that your shoes have become wet."), the second category comprised fear-related situations (e.g. "You have been out with friends. When you return home at midnight, you realize that your door is open."), the third category contained ambiguous fear- and disgust-related situations ("You swim in a big lake. Suddenly, something touches you at your foot.") and the fourth category contained emotionally-neutral situations (e.g. "On a beautiful summer day, you and your friends take the bikes to a lake. You have a picnic."). All paragraphs were validated in an internet-based pilot study with 205 participants (144 females, 59 males, $M_{age}$ = 26.36, $SD_{age}$ = 7.54, $R_{age}$ = 18–84), who were asked about how fearful and disgusting each paragraph was on an analog scale from 0 to 100 (0 = not at all disgusting/fearful, 100 = very disgusting/fearful). For the disgust category, the eight paragraphs with high disgust and low fear ratings ($M_D$ = 41.49, $SD_D$ = 8.97, $M_F$ = 3.88, $SD_F$ = 3.09) were used. The fear category contained eight paragraphs with high fear and low disgust ratings ($M_D$ = 3.71, $SD_D$ = 2.28, $M_F$: $M$ = 45.29, $SD_F$ = 12.55). The ambiguous disgust-fear category contained eight paragraphs with the smallest difference between fear and disgust ratings ($M_D$ = 37.83, $SD_D$ = 19.33, $M_F$ = 35.74, $SD_F$ = 19.77) and the neutral category contained four paragraphs with low ratings of fear and concurrently low ratings of disgust ($M_D$ = .20, $SD_D$ = .24, $M_F$ = .80, $SD_F$ = .45). Overall, 28 text paragraphs were included in the experiment and presented to each participant randomly twice across two blocks. All text paragraphs and their fear and disgust ratings can be requested for use in further studies by the corresponding author. During the experiment, the text paragraphs were presented in a 0.7 cm-high black font on a white, 36 cm-wide x 27 cm-high screen. For emotional deletion, an auditory explanation of how to build a wooden cot was presented at the end of block 1. The MATLAB-based PsychToolbox [34–36] was used to run the experiment on a PC with a 19-inch Version Master Pro 454 monitor. The responses were recorded over a LiTong RTBox response box [37].

## Experimental design and procedures

In this experiment, a 2x4x2 mixed-model design was used with the between-subject factor movie condition (fear and disgust) and the within-subject factors text condition (fear, disgust, fear-disgust, and neutral) and block (1 and 2). Participants were assigned to one of the two

movie conditions stratified by their trait level of *contamination fear* (PI washing score) and their gender and they watched two movies of the same emotion in randomized order. The participants were tested in a small dimmed laboratory room and were seated in front of the monitor at a distance of approximately 50 cm. The instructions were read out loud to the participants by the instructor and were also provided again in written form. They were told to keep the headphones on and keep their arms and hands still for the time of the experiment. After the participants had read the instruction, they could start the movie by pressing a specific key. According to the randomized assigned emotion condition, participants watched either the first disgust or the first fear movie. Following the movie, the participants were asked to rate "How intense did you experience disgust (or fear respectively) while watching the movie scene?" on a seven-point Likert-type scale (0 = not at all intense; 7 = very intense).

Afterwards, the text paragraphs were presented. Each trial begun with a black fixation cross for 500 ms. At the offset of the fixation cross, the text paragraph describing the everyday life situation was presented for 10 s. Afterwards, participants were asked to rate "How intense was your emotional experience while you imagined this situation?" on a seven-point Likert-type scale (0 = not at all; 7 = very intensive). After participants had pressed the button to register the answer, a white blank screen was presented for four seconds and the next trial began. Following all 28 text paragraphs, participants were given the emotion deletion instruction on the screen (see Deletion Task). Subsequently, the second experimental block started with the presentation of the second movie. The procedure of the second block was identical to the first block, aside from the fact that the text paragraphs were presented in a different randomized order and there was no emotion deletion task after the second block. The duration of this experiment was approximately 30 minutes. The experimental procedure is presented in Fig 1.

**Deletion task.** In order to reduce emotional arousal after block 1, a deletion task was presented between the two blocks. The deletion task was only performed after block 1, to minimize carry over effects for the emotion induction in block 2. During the deletion task, participants were instructed to listen carefully to an audio instruction and count how often the word "and" was spoken out loud. They were also instructed to keep their arms and hands still and not to use their hands for counting. Following the instruction, the screen turned black and an audio instruction how to build a wooden cabin was played for 3 minutes. After the audio was finished, participants were asked "how many times was the word 'and' spoken out loud?" They could answer the question by selecting the correct answer out of four given numbers. Subsequently, the answer was logged in by pressing a key.

## Statistical analysis

Data was analyzed twice using ANOVAs with repeated measures as well as Linear Mixed-Effects Models Analysis (LME), whereby the results of the two analyses were expected to converge. All data analyses were applied in R [38]. LME was performed because multiple observations were collected from each participant. In a recent article [39], Aarts et al. found that

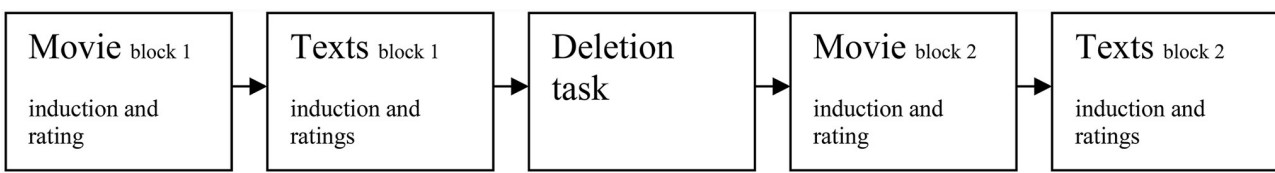

**Fig 1. Experimental procedure.**

multilevel analysis was advantageous for such nested designs. The authors found that the α-error rapidly increases when the average number of observations per cluster increases. Furthermore, the α-error was almost doubled when the inter-cluster correlation increased from 0.1 to 0.5. Because multilevel modeling is not susceptible to the type I error, it was applied in the present study. For the LME analyses, the R-based package lme4 [40] was used. The *p*-values were obtained by likelihood ratio tests, testing hierarchically a more complex model against a less complex model. A variable number of fixed and random effect variables sum up to a GLMM model. The variables are selected by the theoretical assumptions tested in the present study. The intercepts of *participants*, *text paragraph number and movie number* were included in the model as random variables based on the need to use the maximal random effect structure [41]. *Text category*, *block* and *contamination fear* (PI washing score) were selected as fixed effect variables corresponding with the hypotheses. A visual inspection of the dependent variable intensity ratings confirmed the normality assumption. All best-fitted models were controlled for *depression* (BDI-II) and *trait anxiety* (STAI-T). Bayesian T-Tests and ANOVAs were calculated with JASP Version 0.9. [42]. $BF_{10}$ (Bayes Factor) reports the likelihood ratio of the posterior probability of the alternative model ($H_1$) given the data against the posterior probability of the null model ($H_0$) given the data The Bayes factor is commonly interpreted as evidence towards or against one of two competing models. Values between 0.33 and are typically interpreted as weak or anecdotal evidence, values between 3 and 10 as moderate evidence for the alternative hypothesis (or between 0.1 and 0.33 as moderate evidence for the null hypothesis) and values larger 10 (or < 0.1) as strong evidence the alternative (or null) hypothesis. The Bayes factor can also be understood as a quantification of the change in beliefs about the relative plausibility of the competing hypotheses on the basis of the observed data. Unlike the *p*-value, the Bayes Factor also has an asymptotic property, i.e. with an increasing number of cases *N*, the Bayes Factor strives for either zero or infinity. Because the principles of Bayesian statistics are yet not familiar to all readers, both, frequentist and Bayesian parameters are reported. All data is published under http://dx.doi.org/10.17632/cw8bp2xvtv.3.

## Results

### Demographics and individual characteristics

Participants were assigned to the movie conditions using *contamination fear* (PI washing score) and gender as stratification variables, which is why the two groups do not significantly differ in *trait contamination fear* (PI washing score), $M_F = 4.21$, $SD_F = 3.06$, $M_D = 3.8$, $SD_D = 2.62$, $t(57) = -.55$, $p = .59$, $BF_{10} = .30$. The overall number of participants in the fear movie group was 29, while in the disgust movie group it was 30. There was also no significant difference in the gender frequency, with 26 female and 3 (fear) or 4 (disgust) male participants in each group. In Table 1, all characteristics of participants in general and separated by group for *trait contamination fear* (PI washing score), depression (Beck Depression Inventory) and *trait anxiety* (State-Trait Anxiety Inventory, trait scale) are listed. As explained in the method section, all demographic data was collected before the randomization to the two conditions. As presented in Table 1, participants significantly differed in terms of BDI-II score, which was significantly higher in the fear movie group compared with the disgust movie condition, $D = 2.59$, $t(57) = 2.41$, $p = .02$, $BF_{10} = 2.86$, and STAI-T score, which was significantly higher in the fear movie group compared with the disgust movie condition, $D = 2.59$, $t(57) = 2.32$, $p = .02$, $BF_{10} = 2.47$. Therefore, all following analyses were statistically controlled for the BDI and STAI-T differences.

**Table 1. Characteristics of the participants in the two movie conditions fear and disgust.**

| | Overall ($N = 59$) | | | Fear Movie ($N = 29$) | | Disgust Movie ($N = 30$) | | | | |
|---|---|---|---|---|---|---|---|---|---|---|
| Scale | $M$ | $SD$ | Range | $M$ | $SD$ | $M$ | $SD$ | $t(57)$ | $p$ | $BF_{10}$ |
| Age | 21.54 | 6.31 | 18–60 | 21.59 | 7.42 | 21.50 | 5.01 | .05 | .96 | .27 |
| contamination fear | 4.00 | 2.81 | 0–11 | 4.21 | 3.80 | 3.010 | 2.57 | .55 | .59 | .30 |
| depression | 5.37 | 4.25 | 0–17 | 6.69 | 4.51 | 4.100 | 3.54 | 2.41 | .02 | 2.86 |
| trait anxiety | 36.24 | 9.54 | 5–45 | 39.10 | 10.68 | 33.46 | 7.29 | 2.32 | .02 | 2.47 |

contamination fear = Padua Inventory, sub-scale washing and cleaning (PI Washing); depression = Beck Depression Inventory (BDI-II); trait anxiety = Stait-Trait-Anxiety Inventory, Trait Score (STAI-T)

## Behavioral results

**Emotion induction by movies (manipulation check).** We performed a 2x2x2 ANOVA, with the between-factor *movie emotion* (disgust, fear), the within-factor *movie number* (movie 1, movie 2) and the within-factor *block* (block 1, block 2) with repeated measures across *block* and *movie number* and the dependent variable emotional experience. There was a main effect for *movie emotion*, $F(1,55) = 4.29$, $p = .04$, $\eta^2 = .07$, $BF_{10} = 1.50$, a main effect for *block*, $F(1, 55) = 8.47$, $p < .01$, $\eta^2 = .11$, $BF_{10} = 4.45$, and a statistically significant interaction of *movie emotion* and *movie number*, $F(1, 55) = 9.93$, $p < .01$, $\eta^2 = .13$, $BF_{10} = 5.05$, while the other main effects and interactions did not become significant ($p > 0.1$, $BF_{10} < 1$). For the main effect of *movie emotion*, disgust experience after the two disgust movies was on average 5.85 ($SD_D = 1.46$, $N_D = 30$) and was therefore significantly stronger than the overall fear experience after watching the two fear movies, $M_F = 5.19$, $SD_F = 1.46$, $N_F = 29$; $t(57) = -2.06$, $p = .04$, $d = .45$, $BF_{10} = 2.77$. The main effect *block* can be explained by an overall increase of emotional intensity in the second block, $M = 5.80$, $SD = 1.21$, compared with the first block, $M = 5.25$, $SD = 1.69$, $BF_{10} = 3.97$. The interaction between *movie emotion* and *movie number* can be differentiated by the emotional intensity experienced within the disgust movie group. The scene from the movie *Trainspotting*, $M_{TS} = 6.13$, $SD_{TS} = 1.15$, was experienced more intensely than the *Pink Flamingo* clip, $M_{PF} = 5.57$, $SD_{PF} = 1.67$; $t(29) = 2.17$, $p = .04$, $d = -.40$, $BF_{10} = .68$, while no difference was found within the fear movie condition (*Marathon Man*: $M_{MM} = 4.93$, $SD_{MM} = 1.41$; *Halloween*: $M_{HW} = 5.45$, $SD_{HW} = 1.45$; $t(28) = -1.68$, $p = .11$, $d = .36$, $BF_{10} = .57$). Due to the disgust difference, *movie number* was integrated as a further random effect in the following models.

**Text paragraphs.** In a 4x2 ANOVA, with the within-factor *text category* (disgust, fear, fear-disgust, neutral) and the within-factor *block* (block 1, block 2) with repeated measures across *block* and *text category* and the dependent variable emotional intensity, there was a main effect of *text category*, $F(3, 171) = 56.35$, $p < .01$, $\eta^2 = .48$, $BF_{10} > 100$, while the main effect for *block* ($BF_{10} = .17$) and interaction ($BF_{10} = .03$) did not become significant ($p > .5$). The main effect *text category* is driven by the differences between the intensity ratings to fear situations, $M_F = 4.77$, $SD_F = 1.51$, in comparison with fear-disgust situations, $M_{FD} = 4.59$, $SD_{FD} = 1.70$, and in comparison with disgust situations, $M_D = 3.86$, $SD_D = 1.55$, as well as in comparison with neutral text situations, $M_N = 3.49$, $SD_N = 1.86$. To test whether the emotional intensities differed between these four emotional text categories, six paired t-tests were performed to cover all combinations of text conditions. The α-level was adjusted (α = .008) using the Bonferroni correction, whereby all paired t-tests became significant, $p < .008$, $BF_{10} > 3$, thus supporting the distinct nature of the four text categories.

**Hypothesis 1: Bias following disgust and fear induction.** An LME was performed to analyze the contribution of *movie emotion* (fear, disgust), *text category* (fear, fear-disgust, disgust, neutral) and *trait contamination fear* (PI washing score; points) in predicting emotional intensity as explained in the method section. The forward model selection (Table 2) resulted in a best-fitted model comprising a first interaction of *movie emotion* and *text category* and a second interaction of *text category* and *trait contamination fear* (PI washing score) in explaining the dependent variable experienced emotional intensity, $AIC = 11074$; $\chi^2(2) = 42.52$, p < .01.

The first interaction is driven by the different emotional experience after watching one of the two movies. After watching a disgust movie, the reported emotional intensity in response to disgust-related text categories, $M_D = 3.98$, $SD_D = 1.56$, $M_{FD} = 4.68$, $SD_{FD} = 1.72$, was higher compared with the reported emotional intensity after watching a fear movie, $M_D = 3.73$, $SD_D = 1.52$, $M_{FD} = 4.50$, $SD_{FD} = 1.67$. However, the intensity difference between the two movie conditions was not significant ($z = .74$, $p = .46$) between disgust and fear-disgust text paragraphs. After watching a fear movie, the reported emotional intensity in response to fear was higher compared with the reported emotional intensity after watching a disgust movie, $M_F = 4.65$, $SD_F = 1.61$, $M_N = 3.34$, $SD_N = 1.86$. This intensity difference between the two movie conditions was significantly different for fear ($z = 4.71$, $p < .01$) and neutral ($z = 4.37$, $p < .01$) compared with disgust text paragraphs. These results are pictured in Fig 2.

**Hypothesis 2: Stronger disgust sensitivity in participants with higher PI washing scores.** The second significant interaction in this best-fitted model is the interaction of the *text category* and *PI washing score*, which is illustrated in Fig 3. The interaction is driven by the increased emotional intensity experienced in disgust-related situations for people with higher *trait contamination fear* (PI washing score). Thus, the experienced emotional intensity to disgust situations increased significantly more strongly, $a_0 = 3.40$; $b_x = .11$, compared with fear-disgust situations, $a_0 = 4.32$; $b_x = .05$, $z = -2.59$, $p = .01$, while on the other hand the experienced emotional intensity to fear situations, $a_0 = 4.77$; $b_x = -.01$ (compared with disgust: $z = -5.58$, $p < .001$), and neutral situations decreased, $a_0 = 3.58$; $b_x = -.04$ (compared with disgust: $z = -5.57$, $p < .001$), with higher *trait contamination fear* (PI washing score).

**Hypothesis 3: Emotion-specific habituation effect across the two blocks.** In order to test whether the experienced emotional intensity varies across the two blocks in the *movie emotion* and *text category* conditions or if there is an influence of *trait contamination fear* (PI washing score), a more complex saturated model was constructed containing an interaction of *block* and *text category*. The LME analysis tested the complex saturated model against the less complex, best-fitted model (model 3: $AIC = 11072$), which did not result in a significant

**Table 2. Forward model selection.**

| Model | AIC | deviance | $\chi^2$(df) | p |
|---|---|---|---|---|
| text intensity ~ random | 11137 | 11129 | | |
| text intensity ~ PI | 11140 | 11128 | $\chi^2(0) = .87$ | < .01 |
| text intensity ~ Emo | 11135 | 11119 | $\chi^2(2) = 9.03$ | < .01 |
| text intensity ~ Emo * Mov + PI | 11113 | 11087 | $\chi^2(2) = 27.87$ | < .01 |
| text intensity ~ Emo * PI + Mov | 11104 | 11078 | $\chi^2(0) = 8.00$ | < .01 |
| text intensity ~ Emo * PI + Mov * Emo | 11074 | 11042 | $\chi^2(2) = 42.52$ | < .01 |

Displayed are models that explained significantly more variance compared with more simple models. Not displayed are several single and interaction models containing all possible variations of the depicted variables that did not explain significantly more variance.

Text intensity = dependent variable experienced emotional intensity, Emo = *text category*, PI = *trait contamination fear (PI washing score)*, Mov = *movie emotion*, AIC = Akaike information criterion

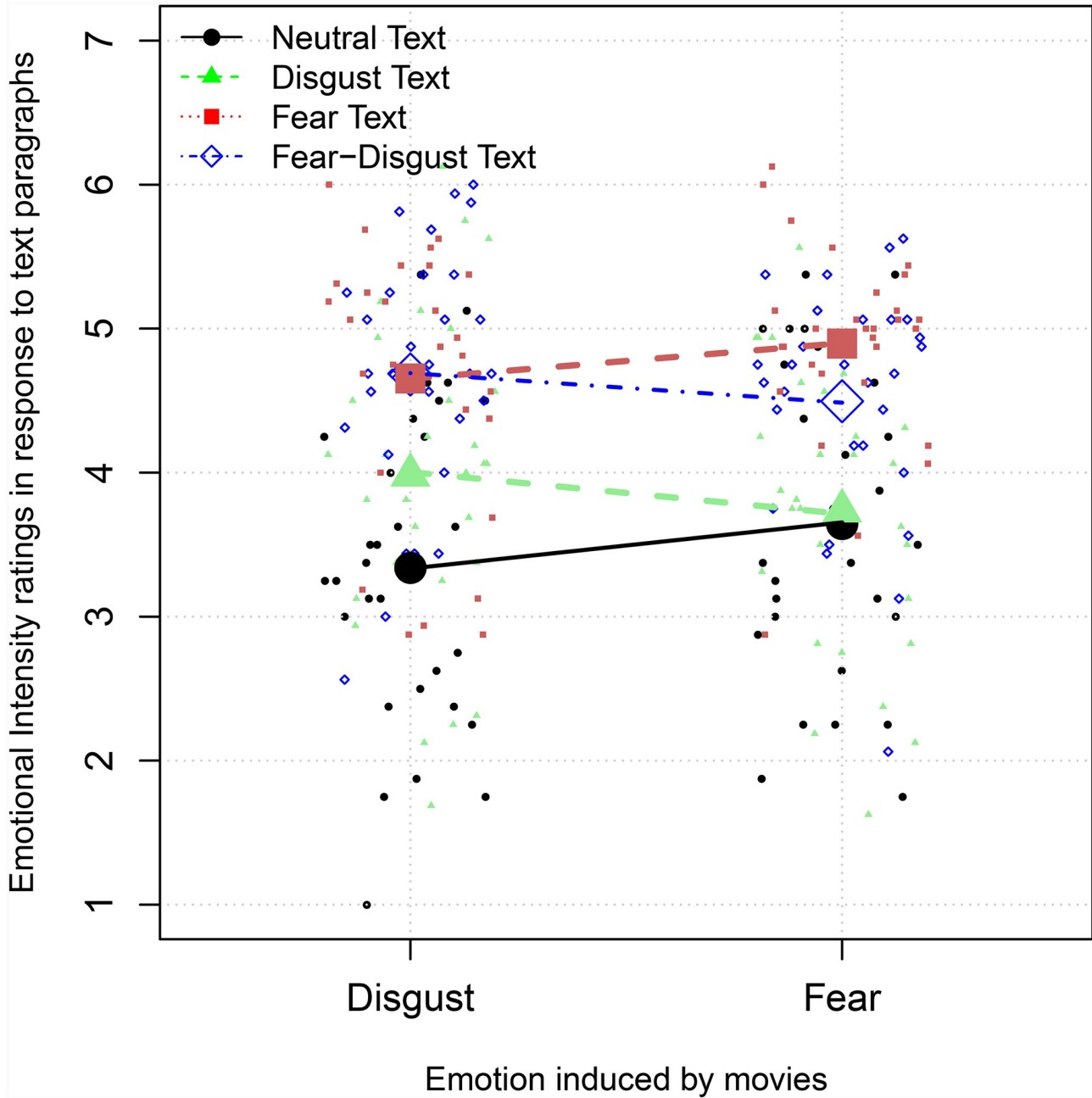

**Fig 2. Movie biases.** Modeled predictions (lines) of the reported emotional intensity to the situations described in the text paragraphs for each participant (small dots) and for both movie conditions (big dots: disgust and fear movies).

amount of more explained variance (model 6: $AIC$ = 11139; $\chi^2(3)$ = 1.12, p = .77). Therefore, we did not observe a significant effect of *block* on the experienced emotional intensity as proposed in hypothesis 3.

## Discussion

The aim of the present study is to better understand the extent to which disgust- and fear-specific state and trait factors influence the interpretation of emotional-ambiguous situations.

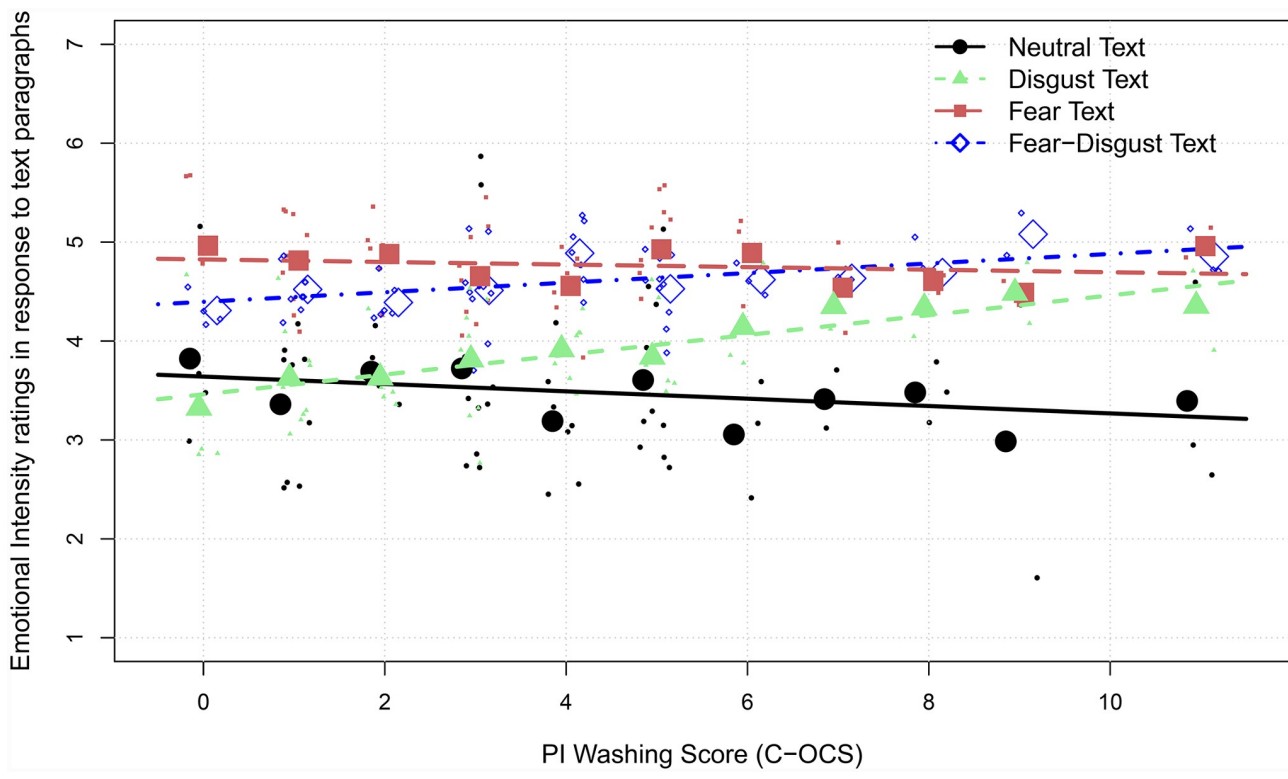

**Fig 3. Trait biases.** Modeled predictions (lines) of the reported emotional intensity in response to the situations described in the text paragraphs for each PI washing score (large dots) and for each participant (small dots). The data was corrected for the random effects of participant and text condition.

There are two major findings in this experiment. First, there is evidence of a state-dependent emotion-specific interpretation bias. Emotional intensity after reading fear- or disgust-specific or ambiguous vignettes describing everyday situations were differentially influenced by whether participants had watched a fear- or disgust-inducing movie immediately beforehand. The text vignettes were experienced as emotionally more intense after watching a movie inducing a congruent emotion. The second finding supports evidence of a connection between higher trait contamination fear and emotional experience in disgust-related situations. Disgust-specific text paragraphs were particularly interpreted as more intense by subjects with higher trait contamination fear. Even though this relationship was not amplified by any movie induction, it supports a model in which the intensity of disgust experience is influenced by individual traits (disgust sensitivity) and situational variables (previously experienced emotions). Contrary to hypothesis 3, there was no habituation of disgust and fear between the two experimental blocks.

The first main finding supports hypothesis 1, which postulated that previous experience of disgust and fear can induce an emotion-specific interpretation bias. This broadens earlier results that do not differentiate between disgust and fear in the outcome variables after inducing a disgust- and fear-specific interpretation bias [12]. Because the disgust and fear induced by the text paragraphs were validated in a large sample and text vignettes were selected by their disgust- and fear-specificity, the independently-measured outcome variable of "emotional intensity" was interpreted as category-congruent emotion-specific arousal. Initially watching a fear movie was followed by increasing emotional experience in fear-related and neutral everyday life situations compared with the disgust induction. Across all blocks and groups, the

strongest response was after fear paragraphs, which support an overall activation [3]. The fear-specific activation of emotional intensity particularly in neutral situations can be seen as evidence of the generalizing nature of fear. On the other hand, having experienced disgust was followed by stronger emotional intensity in disgust-related and disgust-fear-ambiguous everyday situations. These findings can be interpreted using embodiment theories and emotion specific cognitions. Niedenthal [43] postulated that the comprehension of sentences with emotional meanings (e.g. the texts vignettes) is facilitated by reenactment of congruent emotional bodily states. These states are of course easier accessible after congruent compared to incongruent emotional induction. The generalization effect of fear induction on neutral situations can be also explained in the context of this account. Oosterwijk, Topper, Rotteveel and Fischer [44] found that general arousal was elevated after fear knowledge activation, which also had a generalizing effect on the neutral condition. Fear appraisals center on the estimation of danger and harm [10] and might enhance the generalization of fear across text vignettes. The disgust-specific effects could be explained by disgust-specific cognitions. Hereby the laws of contagion and similarity [2] result in contamination (law of contagion) of similar disgust-related text objects (law of similarity) through disgust induction.

Contrary to the hypothesis 3, no disgust or fear habituation between the two experimental blocks occurred. One possible reason for the missing habituation is the emotion deletion task after the first block, which should be excluded in future replications. Another potential reason is the implementation of two different disgust and two different fear movies, which might have renewed the arousal and can therefore be seen as further evidence of the strong arousal-based state bias induced by the films. Both groups significantly differed in trait anxiety, still participants in the disgust group experienced stronger emotional intensity to disgust and ambiguous text vignettes compared to participants in the fear group. Furthermore, we statistically controlled for this a priori group difference. Taken together, a fear-induced embodied emotional state resulted in harm-centered cognitions and an overall stronger emotional response accompanied by an emotion-congruent bias, whereby a disgust-induced emotional state resulted in disgust-specific cognitions and a smaller overall emotional response as well as being accompanied by an emotional-congruent bias. These findings of the movie-induced emotional state highlight the importance and strength of the situational context on the emotional evaluation of emotional-ambiguous situations.

The second main finding provides evidence of the association between higher *trait contamination fear* (PI washing score) and stronger disgust experience in disgust-related everyday situations, whereby fear experience was independent of the degree of trait contamination fear, while the response intensity to disgust-related text paragraphs significantly depended on the level of trait contamination fear. Again, the fear response can be theoretically explained by an overall activation as a defense response to threat [3]. On the other hand, the disgust response depended on the level of trait disgust sensitivity (respectively trait contamination fear) [45]. The positive relationship between trait contamination fear and disgust experience can be seen in line with the cost and benefit hypothesis [46], proposing that individuals with lower trait contamination fear experience low activation and can therefore decide if disgust-related situations are threatening or not. In this subgroup disgust is not embodied and an increased use of a wide range of information is available to interpret the ambiguous situations. In individuals with higher trait contamination fear, disgust stimuli are processed with stronger reenactment of emotional bodily states [43], disgust is more strongly embodied, which increases the disgust-specific cognitions of contagion and similarity [2]. Therefore, accessible information is probably more disgust-as-threat related, which amplifies the experienced intensity of disgust. The trait-related influence of trait contamination fear on the experience of disgust-related situation was not further amplified by the different fear and disgust movies, probably due to the

small range of trait contamination fear in the healthy sample of this study. Again, we theoretically postulate that the activation of the fear component is strong in humans in general, while the activation of the disgust component is much more dependent on trait, with a more intense disgust response in people with higher trait contamination fear. These findings highlight the strong mixed-emotional basis of trait contamination fear.

These findings have clinical implications, including in terms of highlighting the strength of context factors in mixed-emotional situations. Because the text vignettes were emotion-congruently influenced by the movies inducing disgust and fear, this finding highlights the suggestible nature of everyday life situations. In line with the proposals of Craske [47] for improving exposure therapy, exposure techniques have to be embedded in a diverse variety of situations to challenge this strong influence of situational emotional-state biases. Moreover, across the levels of C-OC symptoms, people in this sample did not differ in their intensity ratings to the fear-related text vignettes but in their intensity ratings to the disgust-related vignettes. People with high arousal tend to interpret situations more easily in a familiar direction. Due to the strong disgust-specific trait bias, therapeutic techniques should target disgust more directly to reduce the amount of accessible disgust-as-threat information. The results encourage further investigating additional disgust-specific techniques like counter-conditioning and unconditioned stimuli cognitive reevaluation techniques [48,49] as well as treatment aiming to reduce mental contamination [50].

## Limitations

There are several limitations in this study that warrant mentioning here. The main limitation of this study is the non-clinical population, which might have limited the chance of finding co-variations between trait disgust sensitivity and emotional response to the text-based situations. It might also limit the degree to which our results can be generalized to clinical populations. However, previous research [15,51] has found that thoughts and behaviors in OCD differ more in quantitative rather than qualitative aspects from those observed in non-clinical individuals, thus supporting the idea that basic aspects of OCD can be investigated on a continuum between OCD patients and non-clinical individuals. Nevertheless, our findings should be replicated in a clinical C-OCD population. Second, the deletion task after the first block might have been the reason for the missing habituation effect comparing the first and second block and more generally two blocks might be insufficient to register habituation effects. The third limitation of this study is the gender ratio of 89% women. A more gender diverse sample might control for disgust-specific gender effects, taking in account the notion that women tend to be more sensitive to disgust [52,53]. The exact same results were found by calculating all results again excluding the 7 male subjects, therefore our results have to be seen as applying primarily to women. Furthermore, the average age of the subjects is 21.54 years and research [54] shows that emotional reactivity is changing across lifespan. However, the critical age of development of obsessive-compulsive disorders is between 20 and 30 years, which is why a focus of research on this age group seems justified in this context. A fourth limitation is that no neutral movie condition was included in this experiment, which questions the emotion specificity. Nonetheless, because the movies influenced the response to the emotion-congruent text paragraphs, the effects found in this experiment can be interpreted as emotion-specific. A fifth limititation of the study are the different durations of the movies, which potentially lead to different levels of emotional evocation, or different duration of evocation. In future studies movie clips with a more similar length should be used. Furthermore, only a small number of video clips were used.

## Conclusions

This study has investigated the extent to which disgust- and fear-specific state and trait factors influence the experience of emotional-ambiguous situations. There are three major implications. First, the developed and validated new set of text paragraphs for disgust and fear induction contributes to future disgust and fear research. Second, the fear and disgust components of an emotional response to mixed-emotional situations are strongly influenced by the situational context. While the fear context biases embodied information processing and emotional experience across all presented situations, the disgust context bias was more specific for emotion-congruent situations. This finding highlights the strength of situational context on interpretation bias for mixed-emotional disorders. Third, across the levels of trait contamination fear, participants did not differ in their fear experiences to everyday situations (which was overall strong) but in their disgust experiences. This result supports the important role of disgust for C-OCD and further emphasizes the importance of improving C-OCD related therapeutic approaches in reducing strong disgust experience.

## Supporting information

**S1 Appendix.**
(PDF)

## Acknowledgments

We express our appreciation to the individuals who participated in the experiment. The authors further thank Friederike Degwitz for her assistance in participants' assessment. The content of this article partly overlaps with the authors' thesis (Fink J. Differenzierung von Ekel und Angst und therapeutische Maßnahmen zur Ekelreduktion am Bei-spiel von kontaminationsbezogenen Zwangsstörungen [dissertation]. Leipzig, Germany: Universität Leipzig; 2018.).

## Author Contributions

**Conceptualization:** Jakob Fink-Lamotte, Andreas Widmann, Judith Fader, Cornelia Exner.

**Data curation:** Jakob Fink-Lamotte, Andreas Widmann.

**Formal analysis:** Jakob Fink-Lamotte.

**Investigation:** Jakob Fink-Lamotte, Judith Fader.

**Methodology:** Jakob Fink-Lamotte, Andreas Widmann, Judith Fader.

**Project administration:** Jakob Fink-Lamotte.

**Software:** Andreas Widmann.

**Supervision:** Cornelia Exner.

**Visualization:** Jakob Fink-Lamotte.

**Writing – original draft:** Jakob Fink-Lamotte.

**Writing – review & editing:** Andreas Widmann, Judith Fader, Cornelia Exner.

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
