## [Decision Letter · Decision Letter 0]

20 Mar 2020

PONE-D-20-03172

Interpretation bias and contamination-based obsessive-compulsive symptoms influence emotional intensity related to disgust and fear

PLOS ONE

Dear Dr. Fink,

Thank you for submitting your manuscript to PLOS ONE. After careful consideration, we feel that it has merit but does not fully meet PLOS ONE’s publication criteria as it currently stands. Therefore, we invite you to submit a revised version of the manuscript that addresses the points raised during the review process.

We would appreciate receiving your revised manuscript by May 04 2020 11:59PM. To enhance the reproducibility of your results, we recommend that if applicable you deposit your laboratory protocols in protocols.io, where a protocol can be assigned its own identifier (DOI) such that it can be cited independently in the future. For instructions see: http://journals.plos.org/plosone/s/submission-guidelines#loc-laboratory-protocols

We look forward to receiving your revised manuscript.

Kind regards,

Zezhi Li, Ph.D., M.D.

Academic Editor

PLOS ONE

Journal Requirements:

2. Please provide additional details regarding participant consent. In the Methods section, please ensure that you have specified (1) whether consent was informed and (2) what type you obtained (for instance, written or verbal). If your study included minors, state whether you obtained consent from parents or guardians. If the need for consent was waived by the ethics committee, please include this information.

Reviewers' comments:

Reviewer's Responses to Questions

**Comments to the Author**

1. Is the manuscript technically sound, and do the data support the conclusions?

Reviewer #1: Partly

Reviewer #2: Yes

2. Has the statistical analysis been performed appropriately and rigorously? 

Reviewer #1: Yes

Reviewer #2: Yes

3. Have the authors made all data underlying the findings in their manuscript fully available?

Reviewer #1: Yes

Reviewer #2: Yes

4. Is the manuscript presented in an intelligible fashion and written in standard English?

Reviewer #1: Yes

Reviewer #2: Yes

5. Review Comments to the Author

Reviewer #1: Dear author, I really enjoyed reading the manuscript and as you will see, I believe that some elements still need to be addressed that would make the claims presented in this manuscript clearer. Please find here next a series of comments on a point by point basis.

p.5. Please clarify the ambiguity of the sentence "Found that ambiguous fear-disgust stimuli, like fear stimuli, were rated as more fearful after disgust...". "as well as..." is preferable.

Materials and methods

p.6. Please check language. "will Be interact" should be "will interact"

p.6. Considering the known sexual dimorphism related to the neurobiology of fear processing (e.g., Lebron-Milad, K., et al. (2012)), Can the authors justify the inclusion of only a 11% male participants instead of only assessing females (or trying to assess a more balanced number of participants of either sex?).

p6. Please correct all reported statistical values in the manuscript and report rounded decimals to two.

p7. What do the authors consider to be "relatively high"? Please report M and SD of trait anxiety scores of both groups.

p8. Please briefly describe the Trainspotting movie sequence in the same way it is done for the first movie clip. This is important to show the reader the nature of the disgusting-eliciting material.

p.9.

l.12. Please report that this was presented twice in blocs. Please report the type of presentation (random?).

l.15. Can the authors justify why they had to introduce an emotional debriefing at the end of block 1 and not block 2? Also, emotional debriefing is not necessarily "emotional reduction", therefore the authors should prefer to call this auditory explanation differently and consistent with the "deletion task" as explained later in this section.

p.11. Throughout the manuscript the authors refer to the same task as "debriefing task", "distraction task (fig.1) and "Deletion task". Please adapt this and use the same denomination consistently in the manuscript.

p.12. I appreciate the use of Bayesian statistics. However, the rationale behind as well as the description should be made more intelligible to the reader (of relevance: Dienes, 2014; Wagenmakkers, Verhagen & Ly, 2016).

Furthermore, I think the authors should motivate the presentation of Bayes next to their frequentist equivalent, for instance, in p.12 they report the significant t-test and then report the Bayes factor. It is unclear to the reader why both are reported. This could be shortly motivated in p.12 first paragraph.

Results

See comment about reporting rounded values.

Discussion

p.18. The authors claim that the strongest response is in line with "fast activations".. I believe that it is incorrect to equate magnitude of responses to latency., which was not explicitly assessed anyways. However, considering that the authors used an RTBox response box, if the RT's were collected these could give insight into response latency of either manipulation (and should be reported somewhere either in the ms or supplementary material). For instance, were the evaluations of fear paragraphs faster compared to disgust related ones?

p.18. The authors claim that a bottom-up threat evaluation process is at play in the case of fear. however, the movies last long and the ratings happen after a certain period of time. Therefore, it is very likely that when rating happens ample top-down emotion regulation has taken place, even more so considering that this is written material. I think that it is crucial that the authors try to explain their reasoning behind the generalising aspect of fear further.

Furthermore, it would seem that an embodied process is in line with their results in regards of disgust. Extensive literature (e.g., embodiment, contagion etc.) demonstrates how emotion induction facilitates the processing of congruent emotional information. I do not think that the authors elaborate sufficiently on the mechanisms that underly the emotion induction process and how these could explain the differences between congruency on the one hand and generalisation in the other.

p.19. Check spelling. “Can be could”.

General comments

I enjoyed reading the manuscript and think that the results are very important to understand the dynamics between the processing of fear and disgust. I also appreciated their use of Bayesian statistics and the fact they acknowledged a series of limitations explicitly. I think however, that the authors should, as suggested, elaborate more on their interpretations on what possible mechanisms could underly the differences found between treatments. I also believe that if they collected RT’s these can help greatly in the interpretations.

Please check language/spelling errors.

Reviewer #2: The authors investigated the extent to which disgust- and fear-specific state and trait factors

influence the experience of emotional-ambiguous situations. Using a text paragraph-based interpretation bias paradigm, the authors found that fear and disgust components of an emotional response to mixed-emotional situations are strongly influenced by the situational context, and across the levels of trait contamination fear people did not differ in their fear experiences to everyday situations, but in their disgust experiences. In general, I think the research topic is quite interesting. The manuscript is well written. I only have a few questions.

1. Previous studies have shown that women tend to be more sensitive to disgust, I wander whether the disgust-effects may due to gender effects, given that in the present study, 52 out of 59 subjects are females. Regarding the age, similar question raises, the average age is 21.54 years old with a range of 18-60 years, that means lots of subjects in their early 20s, only a few subjects has an older age. The gender and age effects may have a mixed effect on the observed results.

2. How many subjects participate this study? In Abstract it says 57, however in the Results and Table 1 it seems there are 59 subjects in total. Please clarify the subject numbers.

6. PLOS authors have the option to publish the peer review history of their article (what does this mean?). If published, this will include your full peer review and any attached files.

Reviewer #1: No

Reviewer #2: No

---

## [Author Response · Author response to Decision Letter 0]

27 Mar 2020

1. Journal Requirements:

1.1. Please ensure that your manuscript meets PLOS ONE's style re-quirements

We changed the manuscript concerning file naming, main body and title page in line with the Journal requirements.

1.2. Please provide additional details regarding participant consent. In the Methods section, please ensure that you have specified (1) whether consent was informed and (2) what type you obtained (for instance, written or verbal). If your study included minors, state whether you obtained con-sent from parents or guardians. If the need for consent was waived by the ethics committee, please include this information.

In the method section we included the information, that all participants gave their written informed consent (p. 6).

2. Referee 1:

2.1. Please clarify the ambiguity of the sentence "Found that ambiguous fear-disgust stimuli, like fear stimuli, were rated as more fearful after dis-gust...". "as well as..." is preferable.

On p.4 clarified the types of stimuli used in the experiment (“homophone words”) and reported an example (“dye/die”), furthermore we included “as well as” as the reviewer suggested.

Materials and methods

2.2. Please check language. "will Be interact" should be "will interact".

We changed the sentence on p. 5 in the suggested manner and also checked our entire manuscript carefully for spelling and grammatical issues.

2.3. Considering the known sexual dimorphism related to the neurobi-ology of fear processing (e.g., Lebron-Milad, K., et al. (2012)), Can the authors justify the inclusion of only a 11% male participants instead of only assessing females (or trying to assess a more balanced number of participants of either sex?). 

The reviewer is right and makes an important point for disgust research. Unfortunately, this is a limitation to which we did not pay attention during the recruitment of the participants. We already describe this in our limita-tion section: "The third limitation of this study is the gender ratio of 89 % women. A more gender diverse sample might control for disgust-specific gender effects, taking in account the notion that women tend to be more sensitive to disgust (Olatunji, Sawchuk, Arrindell, & Lohr, 2005; Schienle, Schäfer, Stark, Walter, & Vaitl, 2005)” (p.20). Yet, to contribute to the reviewers’ point, we have calculated the results again only with the female subjects and without the 7 men and found exactly the same results. The main results are presented in table X (below), which corresponds to table 2 in our manuscript. In order to point this out we added a further sentence in the limitation section (p. 20): "The exact same results were found by calcu-lating all results again excluding the 7 male subjects, therefore our results have to be seen as applying primarily to women” In the light of the same results, we argue that the gender effect has no significant influence on the results here. We would thus rather keep the men in the sample in order to avoid a power reduction of the statistical analyses, but we have strength-ened this point in the limitation section. However, if the reviewer was to recommend that we exclude the male subjects and repeat all analyses with a females only sample we would be prepared to due so.

Table X. Forward model selection, N =52, female sample

Model AIC devi-ance χ2(df) p

text intensity ~ random 9717 9709 

text intensity ~ Emo 9714 9699 χ2(2) = 10.25 <.01

text intensity ~ Emo * Mov + PI 9693 9667 χ2(2) = 31.37 <.01

text intensity ~ Emo * PI + Mov 9683 9657 χ2(0) = 10.35 <.01

text intensity ~ Emo * PI + Mov * Emo 9648 9616 χ2(2) = 40.61 <.01

Text intensity = dependent variable experienced emotional intensity, Emo = text category, PI = trait

2.4. Please correct all reported statistical values in the manuscript and report rounded decimals to two. 

We corrected all reported statistical values and now report rounded deci-mals of two in the whole manuscript.

2.5. What do the authors consider to be "relatively high"? Please report M and SD of trait anxiety scores of both groups.

We are thankful for this comment by the reviewer. Here, our interpretation was misleading. Because the mean STAI scores showed low to moderate anxiety, anxiety was relatively high for a healthy sample. To make this interpretation more clear, we included the values from the result section and changed the sentence on p. 5 to: “The trait anxiety scores showed low to moderate anxiety in both groups (Fear-group: M = 39.10, SD = 10.68; Disgust-group; M = 33.46, SD = 7.29) and no participant met the criteria for current mental disorders […]”.

2.6. Please briefly describe the Trainspotting movie sequence in the same way it is done for the first movie clip. This is important to show the reader the nature of the disgusting-eliciting material.

To describe the movie scene, we added the sentence „Hereby, a man sits on a very dirty and disgusting toilet and tries to extracts a bag of drugs from his anus” on p. 7.

2.7. Please report that this was presented twice in blocs. Please report the type of presentation (random?).

To address the reviewers’ points, we changed the sentence to “[…] and presented to each participant randomly twice across two blocks” on p. 8.

2.8. Can the authors justify why they had to introduce an emotional debriefing at the end of block 1 and not block 2? Also, emotional debrief-ing is not necessarily "emotional reduction", therefore the authors should prefer to call this auditory explanation differently and consistent with the "deletion task" as explained later in this section.

We introduced the deletion task only after block 1, because we wanted to minimize carry over effects for the second emotional induction (the second movie). After block 2, no further emotional induction was introduced, therefore we did not include another deletion task. However, at the end of the experiment all subjects were asked whether they needed psychological consultations because of the emotional induction. To clarify this point we included the following sentence on p. 9: “The deletion task was only per-formed after block 1, to minimize carry over effects for the emotion induc-tion in block 2.” And, also in line with 1.9., we changed debriefing to dele-tion throughout the manuscript.

2.9. Throughout the manuscript the authors refer to the same task as "debriefing task", "distraction task (fig.1) and "Deletion task". Please adapt this and use the same denomination consistently in the manuscript.

We thank the reviewer for his/her thorough reading. In line with his/her suggestion we changed it consistently throughout the manuscript and in fig. 1 to “deletion task”.

2.10. I appreciate the use of Bayesian statistics. However, the rationale behind as well as the description should be made more intelligible to the reader (of relevance: Dienes, 2014; Wagenmakkers, Verhagen & Ly, 2016). Furthermore, I think the authors should motivate the presentation of Bayes next to their frequentist equivalent, for instance, in p.12 they report the significant t-test and then report the Bayes factor. It is unclear to the reader why both are reported. This could be shortly motivated in p.12 first paragraph.

To make the rational and the descriptions of the Bayes Factor more intelli-gible to the reader we included examples for different Bayes Factors as well as the asymptotic property of BF. Therefore, we included the follow-ing sentences on p. 10f.: “The Bayes factor is commonly interpreted as evidence towards or against one of two competing models. Values between 0.33 and are typically interpreted as weak or anecdotal evidence, values between 3 and 10 as moderate evidence for the alternative hypothesis (or between 0.1 and 0.33 as moderate evidence for the null hypothesis) and values larger 10 (or < 0.1) as strong evidence the alternative (or null) hy-pothesis. The Bayes factor can also be understood as a quantification of the change in beliefs about the relative plausibility of the competing hy-potheses on the basis of the observed data. Unlike the p-value, the Bayes Factor also has an asymptotic property, i.e. with an increasing number of cases N, the Bayes Factor strives for either zero or infinity”. 

Because some readers will be unfamiliar with Bayesian statistics, we still report both, frequentist and Bayesian statistics. To make this clear we included the sentence “Because the principles of Bayesian statistics are yet not familiar to all readers, both, frequentist and Bayesian parameters are reported” on p. 11.

Discussion

2.11. The authors claim that the strongest response is in line with "fast activations". I believe that it is incorrect to equate magnitude of responses to latency., which was not explicitly assessed anyways. However, consider-ing that the authors used an RTBox response box, if the RT's were collected these could give insight into response latency of either manipulation (and should be reported somewhere either in the ms or supplementary material). For instance, were the evaluations of fear paragraphs faster compared to disgust related ones?

Unfortunately, the RTs were not recorded. We would argue that the RTs of evaluations with Likert scales, which were simply collected in this way, do not really provide results that can be meaningfully discussed. This proba-bly is influenced by too many confounded variables (How well do the re-spondents know the answer format? How willing are they to make deci-sions?). The argumentation with "fast activation" is based on the compre-hensive theory on emotions by Bradley et al (2001). However, the reviewer is right, this could be misleading. For this reason, we have deleted the word "fast" (now on p.17) and would assume an "overall activation" in the sense of Bradley, which can also be better derived from the magnitude/intensity of the fear response. In this context, we have deleted the word "fast" from the entire discussion, but we still maintain the assumption that it is an over-all strong activation in the case of fear induction.

2.12. p.18. The authors claim that a bottom-up threat evaluation process is at play in the case of fear. however, the movies last long and the ratings happen after a certain period of time. Therefore, it is very likely that when rating happens ample top-down emotion regulation has taken place, even more so considering that this is written material. I think that it is crucial that the authors try to explain their reasoning behind the generalising as-pect of fear further.

The reviewer makes a good point here. Indeed, research on emotion regula-tion shows that top-down regulation plays an important role in anxiety (e.g. Cisler et al., 2010) and we cannot prove a "bottom-up threat evaluation process" (now p.19) with our methodology. We have therefore decided to delete the argumentation on the "bottom-up threat evaluation process" on p. 17f. In the reviewer's sense, we have replaced the explanations by an elaboration about underlying mechanisms, which is pointed out in the next suggestion of the reviewer (point 1.13.).

2.13. Furthermore, it would seem that an embodied process is in line with their results in regards of disgust. Extensive literature (e.g., embodiment, contagion etc.) demonstrates how emotion induction facilitates the pro-cessing of congruent emotional information. I do not think that the authors elaborate sufficiently on the mechanisms that underly the emotion induc-tion process and how these could explain the differences between congru-ency on the one hand and generalisation in the other.

What we can show is that neutral and fear-related texts are experienced more intensively due to the fear induction compared to the disgust induc-tion. This speaks for a general fear sensitization and fear activation even after a longer fear movie induction and delayed ratings (see 1.11.) But the reviewer may be right, embodied processes can explain the strong disgust experience after disgust induction and the strong fear experience after fear induction. On the basis of our methodology, this explanation is at least more comprehensible than "bottom up processes", which is why we have explained the congruent emotion effects and the generalization effects of fear with the help of embodiment theory (Niedenthal, 2007; Oosterwijk et al., 2010). In contrast, we have explained the disgust-specific effects, in the sense of the reviewer, with the disgust-specific cognitions (law of con-tagion and similarity, Rozin & Fallon, 1987). We have therefore added a corresponding paragraph on p. 17: “These findings can be interpreted us-ing embodiment theories and emotion specific cognitions. Niedenthal (2007) postulated that the comprehension of sentences with emotional meanings (e.g. the texts vignettes) is facilitated by reenactment of congru-ent emotional bodily states. These states are of course easier accessible after congruent compared to incongruent emotional induction. The gener-alization effect of fear induction on neutral situations can be also ex-plained in the context of this account. Oosterwijk, Topper, Rotteveel and Fischer (2010) found that general arousal was elevated after fear knowledge activation, which also had a generalizing effect on the neutral condition. Fear appraisals center on the estimation of danger and harm (Cisler, Olattunji & Lohr, 2009) and might enhance the generalization of fear across text vignettes. The disgust-specific effects could be explained by disgust-specific cognitions. Hereby the laws of contagion and similarity (Rozin & Fallon, 1987) result in contamination (law of contagion) of simi-lar disgust-related text objects (law of similarity) through disgust induc-tion.”

Furthermore, the effects of disgust trait are explained in the same manner on p. 19: “[…] can be seen in line with the cost and benefit hy-pothesis (Carretié, Ruiz-Padial, López-Martín, & Albert, 2011), proposing that individuals with lower trait contamination fear experience low activa-tion and can therefore decide if disgust-related situations are threatening or not. In this subgroup disgust is not embodied and an increased use of a wide range of information is available to interpret the ambiguous situa-tions. In individuals with higher trait contamination fear, disgust stimuli are processed with stronger reenactment of emotional bodily states (Nie-denthal, 2007), disgust is more strongly embodied, which increases the disgust-specific cognitions of contagion and similarity (Rozin & Fallon, 1987).”

2.14. Check spelling. “Can be could”.

Thank you! The sentence was changed in order to adapt to the reviewers comment 1.13.

3. Referee 2:

3.1. Previous studies have shown that women tend to be more sensitive to disgust, I wander whether the disgust-effects may due to gender effects, given that in the present study, 52 out of 59 subjects are females. Regard-ing the age, similar question raises, the average age is 21.54 years old with a range of 18-60 years, that means lots of subjects in their early 20s, only a few subjects has an older age. The gender and age effects may have a mixed effect on the observed results.

The reviewer is absolutely right and makes a valid point. As we arguing in our response to reviewer 1 (point 1.3) we find the exact same results when eliminating all male subjects from our calculations. In this context we strengthened this point in the limitation section by adding the sentence on p. 20 “The exact same results were found by calculating all results again excluding the 7 male subjects, therefore our results have to be seen as ap-plying primarily to women”. Still we would keep our current presentation of the results by arguing, that the gender effect is not strongly influencing the results. 

This also applies to the age effect, here again the reviewer makes a valid point. Research shows different emotional reactivity over the life span. However, the age between 15 and 25 is critical for the development of OCD. From this point of view, the focus on this age group can be justi-fied. To illustrate this argumentation, we have, in the sense of the reviewer, added the following sentence to the limitation section (p. 20): „Further-more, the average age of the subjects is 21.54 years and research (e.g. Streubel & Kunzmann, 2011) shows that emotional reactivity is changing across lifespan. However, the critical age of development of obsessive-compulsive disorders is between 20 and 30 years, which is why a focus of research on this age group seems justified in this context.“

3.2. How many subjects participate this study? In Abstract it says 57, however in the Results and Table 1 it seems there are 59 subjects in total. Please clarify the subject numbers.

We corrected the mistake in the abstract: 59 subjects participated in this study. We further checked our manuscript carefully for spelling and grammatical issues.

The manuscript has been seen and approved by all authors. The manuscript has not been previously published and has not been submitted elsewhere for publication.

---

## [Decision Letter · Decision Letter 1]

14 Apr 2020

Interpretation bias and contamination-based obsessive-compulsive symptoms influence emotional intensity related to disgust and fear

PONE-D-20-03172R1

Dear Dr. Fink,

We are pleased to inform you that your manuscript has been judged scientifically suitable for publication and will be formally accepted for publication once it complies with all outstanding technical requirements.

With kind regards,

Zezhi Li, Ph.D., M.D.

Academic Editor

PLOS ONE

Reviewers' comments:

Reviewer's Responses to Questions

**Comments to the Author**

1. If the authors have adequately addressed your comments raised in a previous round of review and you feel that this manuscript is now acceptable for publication, you may indicate that here to bypass the “Comments to the Author” section, enter your conflict of interest statement in the “Confidential to Editor” section, and submit your "Accept" recommendation.

Reviewer #1: All comments have been addressed

Reviewer #2: All comments have been addressed

2. Is the manuscript technically sound, and do the data support the conclusions?

Reviewer #1: Yes

Reviewer #2: Yes

3. Has the statistical analysis been performed appropriately and rigorously? 

Reviewer #1: Yes

Reviewer #2: Yes

4. Have the authors made all data underlying the findings in their manuscript fully available?

Reviewer #1: Yes

Reviewer #2: Yes

5. Is the manuscript presented in an intelligible fashion and written in standard English?

Reviewer #1: Yes

Reviewer #2: Yes

6. Review Comments to the Author

Reviewer #1: Dear Author,

I see that all previous comments have been addressed satisfactorily.

I appreciate very much how the author elaborated and answered all questions in a very rigorous and sound manner.

Thank you for integrating all my comments in the manuscript and please double check spelling on "extracts" on P.7..

Reviewer #2: The authors have satisfactorily responded to all my questions and made the necessary changes to the manuscript. I do not have any other further questions.

7. PLOS authors have the option to publish the peer review history of their article (what does this mean?). If published, this will include your full peer review and any attached files.

Reviewer #1: No

Reviewer #2: No

---

## [Editor Report · Acceptance letter]

15 Apr 2020

PONE-D-20-03172R1 

Interpretation bias and contamination-based obsessive-compulsive symptoms influence emotional intensity related to disgust and fear 

Dear Dr. Fink:

I am pleased to inform you that your manuscript has been deemed suitable for publication in PLOS ONE. Congratulations! Your manuscript is now with our production department. 

With kind regards,

on behalf of

Dr. Zezhi Li 

Academic Editor

PLOS ONE